# The Association of Dietary Intake with Arterial Stiffness and Vascular Ageing in a Population with Intermediate Cardiovascular Risk—A MARK Study

**DOI:** 10.3390/nu14020244

**Published:** 2022-01-07

**Authors:** Leticia Gómez-Sánchez, Emiliano Rodríguez-Sánchez, Rafel Ramos, Ruth Marti-Lluch, Marta Gómez-Sánchez, Cristina Lugones-Sánchez, Olaya Tamayo-Morales, Ines Llamas-Ramos, Fernando Rigo, Luis García-Ortiz, Manuel A. Gómez-Marcos

**Affiliations:** 1Institute of Biomedical Research of Salamanca (IBSAL), 37007 Salamanca, Spain; emiliano@usal.es (E.R.-S.); martagmzsnchz@gmail.com (M.G.-S.); cristinals@usal.es (C.L.-S.); olayatm@usal.es (O.T.-M.); inesllamas@usal.es (I.L.-R.); lgarciao@usal.es (L.G.-O.); magomez@usal.es (M.A.G.-M.); 2Primary Care Research Unit of Salamanca (APISAL), 37005 Salamanca, Spain; 3Health Service of Castile and Leon (SACyL), 37007 Salamanca, Spain; 4Faculty of Medicine, University of Salamanca, 37007 Salamanca, Spain; 5Unitat of Suport the Recerca of Girona, Institut Universitari d’Investigacio in Atencion Primària Jordi Gol (IDIAP Jordi Gol), 17004 Girona, Spain; rramos.girona.ics@gencat.cat (R.R.); rmarti.girona.ics@gencat.cat (R.M.-L.); 6Institut d’InvestigacioÂ Biomèdica of Girona Dr. Josep Trueta (IDBGI), 17002 Girona, Spain; 7Departament of CiènciesMèdiques, Facultat of Medicina, Universitat of Girona, 17004 Girona, Spain; 8Department of Nursery and Physiotherapy, Faculty of Nursing and Physiotherapy, University Hospital of Salamanca, 37007 Salamanca, Spain; 9Department of Nursing and Physiotherapy, University of Salamanca, 37007 Salamanca, Spain; 10San Agustin Health Center, Illes Balears Health Service (IBSALUT), 07015 Palma of Mallorca, Spain; frigo@ibsalut.caib.es

**Keywords:** dietary patterns, early vascular ageing, intermediate cardiovascular risk

## Abstract

The aim of this study was to analyse the association of diet with arterial stiffness and vascular ageing in a Caucasian population with intermediate cardiovascular risk. We recruited 2475 individuals aged 35–75 years with intermediate cardiovascular risk. Brachial-ankle pulse wave velocity (baPWV) was measured using a VaSera VS-1500^®^ device. Vascular ageing was defined in two steps. Step 1: The 20 individuals who presented kidney disease, peripheral arterial disease, or heart failure were classified as early vascular ageing (EVA). Step 2: The individuals with percentiles by age and sex above the 90th percentile of baPWV among the participants of this study were classified as EVA, and the rest of the individuals were classified as non-EVA. The diet of the participants was analysed with two questionnaires: (1) the diet quality index (DQI) questionnaire and (2) the Mediterranean diet (MD) adherence questionnaire. The mean age of the sample was 61.34 ± 7.70 years, and 61.60% were men. Adherence to the MD was 53.30%. The DQI was 54.90%. Of the entire sample, 10.70% (11.15% of the men and 9.95% of the women) were EVA. In the multiple linear regression analysis, for each additional point in the DQI questionnaire, there was a decrease of −0.081 (95%CI (confidence intervals) −0.105–−0.028) in baPWV; in the MD adherence questionnaire, there was a decrease of −0.052 (95%CI −0141–−0.008). When performing the analysis, separated by sex, the association remained significant in men but not in women. In the logistic regression analysis, there was an increase in MD adherence and a decrease in the probability of presenting EVA, both with the DQI questionnaire (OR (odds ratio) = 0.65; 95%CI 0.50–0.84) and with the MD adherence questionnaire (OR = 0.75; 95%CI 0.58–0.97). In the analysis by sex, the association was only maintained in men (with DQI, OR = 0.54; 95%CI 0.37–0.56) (with MD, OR = 0.72; 95%CI 0.52–0.99). The results of this study suggest that a greater score in the DQI and MD adherence questionnaires is associated with lower arterial stiffness and a lower probability of presenting EVA. In the analysis by sex, this association is only observed in men.

## 1. Introduction

A dietary pattern characterised by a high content of fruits, vegetables, cereals, legumes, walnuts, olive oil, and fish, a low intake of red meat and dairy products, and moderate consumption of alcohol in red wine is related to an improvement in cardiometabolic risk factors and a lower morbimortality of cardiovascular disease, stroke, and cancer [1,2,3,4].

Arterial stiffness evaluated with brachial-ankle pulse wave velocity (baPWV) predicts morbimortality by cardiovascular diseases [5]. Moreover, the relationship between dietary intake and arterial stiffness is not clear. Several studies have shown that certain dietary patterns reduce arterial stiffness [6,7,8,9]. However, other studies have not found such a relationship [10] or have reported a correlation when arterial stiffness was evaluated with the central augmentation index (cAIx), but not when it was assessed with carotid-femoral pulse wave velocity (cfPWV) [7].

Vascular ageing is influenced by arterial stiffness, reflecting the dissociation between the chronological and biological age of the main arteries, with their alteration preceding the appearance of cardiovascular events [11,12,13]. In the last several decades, some epidemiological studies have been conducted to identify the determining factors of vascular ageing, interestingly showing a stronger relationship with morbimortality by cardiovascular diseases than biological ageing [11,13]. The effect of the different dietary patterns on ageing has been analysed in some studies, all of which have reported that calorie restriction and the intake of nutrients with antioxidant and anti-inflammatory effects have a positive impact on ageing [14,15,16,17].

Therefore, the association of different dietary patterns, such as the diet quality index (DQI) [18] and adherence to the Mediterranean diet (MD) [19], with arterial stiffness and early vascular ageing (EVA) is an area of increasing research interest. The hypothesis of the study is that subjects with lower adherence to the DQI and MD questionnaires will have higher arterial stiffness, and a higher percentage of them will be classified as EVA. Thus, considering that the relationship of different dietary patterns with EVA has not been explored in all population groups, the aim of this study was to analyse the association of diet with arterial stiffness and vascular ageing in a Caucasian population with intermediate cardiovascular risk, additionally exploring the presence of differences between sexes.

## 2. Materials and Methods

### 2.1. Study Design

This was a cross-sectional, descriptive, multicentre study framed within a larger research project entitled interMediAte RisK management (MARK) study (registered in ClinicalTrials.gov: NCT01428934) [20].

### 2.2. Study Population

The participants were selected by random sampling among individuals who met the inclusion criteria and visited the doctor’s office in six primary care centres of three Autonomous Communities of Spain between July 2011 and June 2013. A total of 2495 recruited individuals were aged 35–75 years and presented intermediate cardiovascular risk as a 10-year coronary risk between 5% and 15% estimated with the adapted Framingham’s risk equation [21], a 10-year vascular mortality risk between 1% and 5% estimated with the equation for European population [22], or a moderate risk following the criteria of the European Society of Hypertension for the management of arterial hypertension [23]. The exclusion criteria included the following: presenting a disease in the terminal stage, being institutionalised at the time of the visit, or having a history of atherosclerosis. This manuscript presents the analysis of 2475 participants recruited from the MARK study, excluding 20 individuals for not having baPWV measured or for not presenting the DQI or MD adherence questionnaires.

The way in which the clinical data have been collected, the anthropometric measurements, and the analytical variables have been published in the study protocol [20].

### 2.3. Ethics Approval and Consent to Participate

All procedures associated with the study were explained in detail to the individuals who gave their written consent prior to the beginning of the study. This study was revised and approved by the Research Ethics Committee of the Jordi Gol Institute for Primary Care Research, the Drug Research Ethics Committee of Salamanca, and the Ethics Committee of Palma de Mallorca. The principles of the Declaration of Helsinki were followed throughout the entire study [24].

### 2.4. Variables and Measurement Instruments

Prior to the beginning of the study, six healthcare professionals were trained (two in each data-gathering centre) to record the measurements and complete the questionnaires following a standardised protocol.

#### 2.4.1. Assessment of Arterial Stiffness with Brachial-Ankle Pulse Wave Velocity (baPWV)

baPWV was measured using a VaSera VS-1500^®^ device (FukudaDenshi, Denshi, Denshi Co. Ltd., Tokyo, Japan) using an oscillometric method, following the manufacturer’s instructions. One hour before taking the test, the participants were not allowed to smoke or consume caffeine. Moreover, they should come with comfortable clothing and remain at rest for at least 10 min before the measurement. The blood pressure cuffs were adapted to the circumference of the arms and legs. The cuffs were adapted to the arm and leg circumference on both sides. baPWV was calculated using the following equation: baPWV = (0.5934 × height (cm) + 14.4724)/tba (tba is the time interval between the arm and ankle waves) [25]. The median value was used as the cut-off point.

#### 2.4.2. Definition of Individuals with Early Vascular Ageing

Vascular ageing was defined in two steps. Step 1: The 20 individuals who presented kidney disease, peripheral arterial disease, or heart failure, following the criteria established in the 2013 clinical practice guideline of the European Societies of Hypertension and Cardiology for the treatment of arterial hypertension [23], were classified as EVA. Step 2: The individuals with percentiles by age and sex above the 90th percentile of baPWV among the participants of this study were classified as EVA, and the rest of the individuals were classified as non-EVA.

#### 2.4.3. Dietary Assessment

The diet was evaluated with two questionnaires: the diet quality index (DQI) questionnaire and the Mediterranean diet (MD) adherence questionnaire. The DQI questionnaire validated in the Spanish population [18] gathers the frequency of consumption of 18 food groups in the last year, divided into three categories, using standardised rations to quantify the frequency of food intake. The first category includes 8 foods ((i) bread; (ii) vegetables (cooked and raw); (iii) fruits; (iv) milk and yogurt; (v) rice and pasta; (vi) vegetable oils (olive and sunflower); (vii) alcoholic drinks; (viii) cereals (corn flakes, oatmeal, etc.)). This questionnaire records the daily consumption of foods in rations, and, depending on whether the consumption is below, equal to, or over 1 ration/day, the respondent is given 1, 2, or 3 points, respectively, except for the consumption of alcohol, in which case the respondent is given 3 points for the daily consumption of 1 alcoholic drink and 1 point for the consumption of more or less than 1 alcoholic drink per day. The second category includes seven food groups ((i) meat; (ii) sausages; (iii) cheese; (iv) sweets; (v) animal fat (butter, lard); (vi) other vegetable oils (palm oil, etc.); (vii) fast food). This questionnaire records the weekly consumption of foods considered harmful, giving 2 points to the respondent for a consumption between 4 and 6 rations/week and 1 or 3 points for a lower or greater consumption, respectively. The third category includes three food groups: ((i) fish; (ii) legumes; (iii) walnuts)), giving 2 points for a consumption of 2–3 times per week, whereas higher and lower consumptions receive 3 and 1 points, respectively. The total score ranges between 18 and 54 points, with higher scores being associated with the best diet quality. A score ≥39 points are considered a good DQI. The second questionnaire used, i.e., the previously validated MD adherence questionnaire [19], consists of 9 items related to compliance with different aspects of the Mediterranean diet, such as the consumption of olive oil, vegetables, fruits, nuts, and white meat. A score of ≥5 points is considered to indicate good adherence to the Mediterranean diet.

#### 2.4.4. Assessment of Regular Physical Activity

Physical activity was evaluated with the short version of the Minnesota Leisure Time Physical Activity (LTPA) questionnaire validated for the Spanish population in men and in women [26,27]. Interviewers trained in advance collected this questionnaire, which collects detailed information on the physical activity carried out during the previous year, as well as the number of times this activity was carried out and the average duration of each activity in each session. Following the recommendations of the American Heart Association, we classified participants as sedentary if the moderate physical activity carried out was < 675 MET-minutes/week or if the intense physical activity was < 420 MET-minutes/week [28].

#### 2.4.5. Consumption of Alcohol and Tobacco

Using detailed questionnaires, we gathered the type and amount of alcohol consumed and the number of years as a smoker [20]. It was considered that an individual was a smoker if he/she had quit smoking less than one year before the study and as a risk drinker if the consumption of alcohol was ≥140 g/week in women and ≥210 g/week in men.

### 2.5. Statistical Analysis

The results of the continuous variables are shown as mean ± standard deviation and those of the categorical variables as number and percentage. The comparison of means between two independent groups was carried out using a Student’s *t*-test, and the comparison between categorical variables was performed using the χ^2^ test. The association of the scores of the DQI and MD adherence questionnaires with baPWV was assessed through a multiple linear regression analysis, using baPWV as a dependent variable and the scores of the DQI and MD adherence questionnaires as independent variables. Age, sex (0 = woman, 1 = man), years as a smoker, grams of alcohol per week, physical activity performed weekly in MET-minutes/week, and intake of antihypertensive, lipid-lowering, and glucose-lowering drugs (no drug intake = 0; drug intake = 1) were used as adjustment variables. To analyse the association of the indices of the DQI and MD adherence questionnaires with vascular ageing, several logistic regression models were conducted, using *non-EVA* and *EVA* as dependent variables (non-EVA = 0; EVA = 1). The scores of the DQI and MD adherence were used as independent variables (DQI < 39 = 0; DQI ≥ 39 = 1; MD < 5 = 0; MD ≥ 5 = 1). Age, sex (0 = woman, 1 = man), being a smoker, being sedentary, being a risk drinker, and an intake of antihypertensive, lipid-lowering, and glucose-lowering drugs (no drug intake = 0; drug intake = 1) were used as adjustment variables. The analyses were carried out globally and by sex. In the hypothesis test, an α risk of 0.05 was established as the limit of statistical significance. All analyses were performed using the statistical software SPSS for Windows v.25.0 (IBM Corp, Armonk, NY, USA).

## 3. Results

### 3.1. Clinical Characteristics and Vascular Ageing

The general characteristics in terms of lifestyles, cardiovascular risk factors, cardiovascular diseases, and vascular function, globally and by sex, are described in Table 1. Men performed more physical activity and consumed more alcohol and tobacco, whereas the women obtained higher scores in the DQI and MD adherence questionnaires. Men with higher DQI and MD adherence obtained better values of baPWV than those with lower DQI and MD adherence.

The general characteristics, lifestyles, cardiovascular risk factors, and vascular function in the individuals with and without EVA are described in Table 2. The EVA individuals showed lower DQI and MD adherence scores, and a lower percentage of individuals met the adherence criteria. Moreover, they also showed higher values of arterial pressure, triglycerides, glycemia, and creatine. On the other hand, the individuals without EVA presented higher values of total and LDL cholesterol. The results by sex are shown in Appendix A. Only in men were the scores of the DQI and MD adherence questionnaires lower in the individuals with EVA than in those without EVA.

### 3.2. Relationship of Arterial Stiffness with Diet Quality inDex and Mediterranean Diet Adherence

In the multiple linear regression analysis, after controlling for age, sex, and other confounding factors (Table 3), for each additional point in the score of the DQI questionnaire, there was a decrease of −0.081 (95%CI −0.105–0.028) in baPWV; for each additional point in the MD adherence questionnaire, there was a decrease of −0.052 (95%CI −0141–−0.008). When performing the analysis separated by sex, the association was maintained in men but not in women (Table 3).

### 3.3. Association of EVA with Diet Quality Index and Mediterranean Diet Adherence

The logistic regression models, after controlling for possible confounding variables, showed that the increase in MD adherence reduced the probability of presenting EVA, both in the DQI questionnaire (OR = 0.65; 95% CI 0.56–0.97) and in the MD adherence questionnaire (OR = 0.75; 95% CI 0.58–0.97). In the analysis by sex, the association was only maintained in men (DQI: OR = 0.54; 95% CI 0.39–0.76, MD: OR = 0.71; 95% CI 0.52–0.99), as is shown in Figure 1. The associations with the food categories of the DQI questionnaire and the components of the MD adherence questionnaire are shown in Appendix A.

## 4. Discussion

To our knowledge, this is the first study to analyse the relationship of two dietary patterns, evaluated through questionnaires, with arterial stiffness and with the degree of vascular ageing in the Spanish population with intermediate cardiovascular risk, between 35 and 75 years of age. In this sense, adherence to the MD was higher in women. In the multiple regression analysis, arterial stiffness showed an inverse relationship with the scores of the DQI and MD adherence questionnaires after controlling for possible confounding factors. The main result of the study was that, in the logistic regression analysis, the subjects with good DQI and MD adherence had a lower probability of being classified as EVA after controlling for possible confounding variables. However, when performing the analysis by sex, these associations were only maintained in men and varied when the analysis was performed by the categories of the DQI questionnaire and by the different components of the MD adherence questionnaire.

Mean values and the percentage of adherence to the MD was greater in women than in men, which is in line with the results of previous studies [29,30].

The benefits of different heart-healthy dietary patterns were analysed in the *Umbrella Review of Meta-Analyses of Randomized Controlled Trials* carried out by Dinu et al. [31]. In it, they analysed 11 dietary patterns, including, mainly, subjects who were overweight, obese, or diagnosed with type 2 diabetes mellitus. The strongest and most consistent evidence was found for the Mediterranean dietary pattern. This diet showed a reduction in weight, body mass index, blood pressure, insulin glucose, and Hemoglobin A1c (HbA1c) and an improvement in the lipid profile without evidence of potential adverse effects.

In the present work, we found an inverse relationship between arterial stiffness assessed with baPWV and the total scores of the two diet questionnaires analysed. These results are in agreement with those of previous cross-sectional studies, as is shown by the results found in the general population in the EVIDENT study [9] and the data of 291 individuals who had their arterial stiffness evaluated through cfPWV in the EVIDENT 2 study [8]. Both studies found an inverse association of the DQI of the EVIDENT study and the Atlantic diet quality index with arterial stiffness assessed through cfPWV. However, a study that analysed the relationship of three dietary indices with arterial stiffness evaluated through cfPWV and with the central augmentation index in individuals with diabetes mellitus found no association [10]. In the same line, several clinical trials have analysed the effect of different dietary interventions on arterial stiffness or the effect of greater interventions on lifestyles, including diet and physical exercise interventions. The clinical trial that included the largest number of individuals is the NU-AGE study (New Dietary Strategies Addressing the Specific Needs of Elderly Population for Healthy Aging in Europe) [7], which included 1294 participants aged 65 to 79 years, with a 1-year follow-up; the NU-AGE study evaluated, in a sample of 225 individuals, the effects of a dietary intervention based on an MD style, to comply with the dietary recommendations for people over 65 years [32]. They found a between-group change with a central augmentation index of −12.4 (95% CI, −24.4–−0.5; *p* = 0.04) and no differences in cfPWV. In other clinical trials, interventions based on the MD pattern improved the endothelial function in 152 older normotensive adults for six months [33] and in 180 adults with metabolic syndrome after two years [34]. However, a trial conducted in individuals with diabetes mellitus did not find any effects on systolic or diastolic arterial pressure, the central augmentation index, or cfPWV after 12 months of follow-up [35]. Among the trials that have used diet and exercise interventions, individuals with obesity have shown a decrease in cfPWV, baPWV, the central augmentation index, and the central systolic arterial pressure [36]; in a different study performed in hypertensive individuals, the intervention improved carotid stiffness, also observing a regression of atherogenesis in the carotid artery, thereby improving the carotid intima-media thickness [6].

The differences between sexes have been analysed in few studies, although, in the NU-AGE study [7], in contrast with the findings of the present study, the association was only observed in women with the central augmentation index, observing no effect of the intervention on cfPWV. The differences between sexes could be explained by the fact that, although women present higher scores in both questionnaires to evaluate diet quality with respect to men, there are no differences (*p* < 0.05) in the mean value of baPWV between the women who showed good adherence and those who showed poorer adherence; this was not observed in men, since those who showed good adherence had lower values of baPWV than those who showed poorer adherence, and the difference in adherence between individuals with and without EVA is greater. On the other hand, vascular ageing is a gradual process that involves biochemical, enzymatic, and cellular phenomena in the vascular wall. All this, combined with epigenetic and molecular alterations, causes an increase in arterial stiffness and a reduction in compliance that differs according to sex [11,37]. Therefore, sex differences are influenced by hormonal and non-hormonal factors [38,39]. Thus, the protection of endogenous estrogen until menopause in women is well known. In addition, in males, arterial stiffness increases linearly from puberty, indicating that females inherently have stiffer main arteries than males; these effects are mitigated by sex steroids during reproductive life. Height, body fat distribution, and inflammatory factors may also play a role [38,39]. Lastly, it is important to take into account that the measurements of stiffness used evaluates arterial stiffness in different parts of the vascular tree. Thus, cfPWV is a measurement of central arterial stiffness, whereas baPWV assesses peripheral arterial stiffness, and the central augmentation index evaluates both central and peripheral arterial stiffness [40].

Supporting the results described above, we found a relationship between the individuals classified in the EVA group and the scores of the DQI and MD adherence questionnaires. Thus, the individuals classified as EVA obtained lower scores in both questionnaires. Knowledge about the benefits of a healthy diet or dietary pattern on the phenotypes of arterial ageing and the underlying mechanisms is limited [16]. In part, this is probably due to the fact that trials carried out in people with dietary interventions in humans are not easy to conduct because they involve many multiple factors that can contribute to changes in arterial function. It is also difficult to carry out preclinical studies in animals, which limits the knowledge of the effects that diet can have on cells and tissues [13,16]. However, the existing data indicate that diet or dietary patterns based on a high intake of vegetables, fresh fruit, fibre, nuts, and fish and a small intake of processed meats protect the body against arterial ageing [13,41,42]. Although the exact mechanisms through which these dietary patterns modulate arterial ageing on a large scale are still unknown, current knowledge indicates that, in most cases, both oxidative stress and inflammation play an important role [13,16]. For this reason, in recent years, studies have been carried out that analyse the relationship and the mechanisms involved between the components of MD and ageing in animals and people [43]. The study carried out by Cesari et al. [44] found that the daily consumption of several components of MD, such as olive oil, fresh fruits, and vegetables, could inhibit endothelial dysfunction, all of which causes an increase in endothelial progenitor cells and circulating progenitor cells [44]. In 2018, Serino et al. [2] analysed the role that polyphenols had in vascular ageing, finding that polyphenols can help to reduce inflammation by increasing antioxidant capacity, which causes a decrease in vascular stiffness and atherosclerosis and delays vascular ageing. Foods rich in polyphenols include fruits, vegetables, olive oil, and wine. In this line, a study carried out by Ma et al. [45] in three prospective cohorts of US men and women included in the Health Professionals Follow-Up Study and in the NHS (Nurses’ Health Study) with more than two decades of follow-up showed that a higher intake of isoflavones was associated with a moderately lower risk of coronary heart disease. Tofu consumption was also significantly and inversely associated with coronary heart disease risk. The favourable association of tofu was more pronounced in young or postmenopausal women without hormone replacement therapy. These results imply that tofu and other soy products could be incorporated into the diet to facilitate the prevention of ischemic heart disease [45]. However, another study carried out in mice did not find that the consumption of olive oil had a protective effect, either on ageing or on memory, so these beneficial effects of the components of MD may be due to an improvement in the lipid metabolism [46]. Other studies support the hypothesis that the beneficial effects of the consumption of extra virgin olive oil on health can be mediated by its effects on mesenchymal stem cells, which could explain some of the health effects of consuming olive oil, such as the prevention of unwanted ageing processes [44].

The main novelties of this study are as follows: First, it analyses the relationship of the results between two dietary questionnaires with vascular ageing in Caucasian patients with intermediate cardiovascular risk. Second, it shows that subjects with lower adherence to a healthy diet, as assessed by the DQI and MD questionnaires, are more likely to be classified as EVA. Finally, when we analyse the association separately by sex, this is only maintained in men.

To sum up, the findings of this study suggest that healthy dietary patterns are associated with better vascular function and healthier vascular ageing, although their effects differ between sexes. Therefore, further studies should clarify the mechanisms involved in the different components of vascular ageing.

Regarding the main limitations of this study, the characteristics of the cross-sectional analysis do not allow one to infer causality. The results only refer to the Spanish population with intermediate vascular risk and thus may not be generalisable to other groups or ethnicities. Lastly, the evaluation of the intake of foods was conducted self-reportedly by the participants through validated questionnaires; therefore, it is a subjective measure.

## 5. Conclusions

A higher mean score and adherence to the DQI and MD questionnaires are associated with lower values of arterial stiffness and with a decrease in the probability of presenting EVA. When we break down the analysis by sex, the association only holds for men. These results may be useful in clinical practice, helping health professionals to provide healthy dietary recommendations, with the aim of improving vascular ageing, especially in men.

## Figures and Tables

**Figure 1 nutrients-14-00244-f001:**
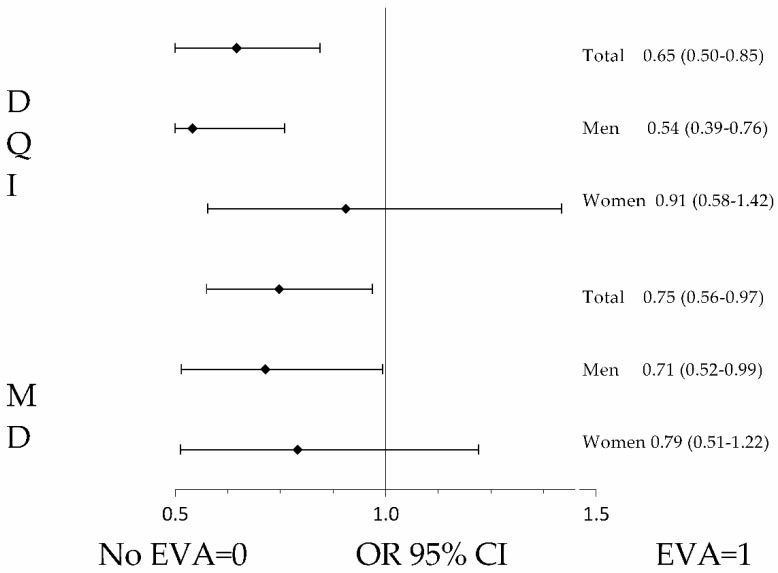
Bars show OR (odds ratio) and 95% CI. Association between arterial ageing and DQI and MD score in the entire sample and by sex. Dependent variable: the presence of early vascular ageing (EVA = 1) versus non-EVA (non-EVA = 0). Independent variables: adherence to DQI and MD (1 = Yes, 0 = No). Adjustment variables: age, sex (0 = woman; 1 = man), being a smoker, being sedentary, being a risk drinker, and an intake of antihypertensive, lipid-lowering, or glucose-lowering drugs (no risk factor or no intake of drugs = 0; having a risk factor or an intake of drugs = 1). DQI: diet quality index; MD, Mediterranean diet; EVA, early vascular ageing.

**Table 1 nutrients-14-00244-t001:** General characteristics of the subjects included in global and by sex.

	Global (2475)		Men (1524)		Women (951)		*p* Value
Lifestyles	Mean or N	SD or (%)	Mean or N	SD or (%)	Mean or N	SD or (%)	
Years of smoker, (years)	31.66	12.82	31.95	12.78	30.74	12.92	0.113
Smoker, n (%)	710	(28.7)	486	(31.9)	224	(23.7)	<0.001
Alcohol, (gr/W)	71.87	116.74	102.01	132.71	23.58	58.86	<0.001
Risk consumption, n (%)	334	(13.5)	284	(18.6)	50	(5.3)	<0.001
Total FA, (METs/m/W)	2462	2495	2864	2815	1817	1683	<0.001
Sedentary, n (%)	2127	(85.9)	1269	(83.3)	858	(90.2)	<0.001
MD, (total score)	5.18	1.73	5.09	1.78	5.32	1.63	0.002
Adherence MD, n (%)	1295	(52.4)	781	51.2)	514	54.0)	0.186
DQI, (total score)	38.71	3.07	38.61	3.15	38.88	2.93	0.028
Adherence DQI, n (%)	1358	(54.9)	803	(52.7)	555	(58.9)	0.005
**Conventional Risk Factors**							
Age, (years)	61.34	7.70	61.11	8.11	61.70	7.00	0.066
SBP, (mmHg)	137.25	17.37	139.09	17.05	134.32	17.48	<0.001
DBP, (mmHg)	84.58	10.23	85.67	10.44	82.84	9.63	<0.001
BP, (mmHg)	52.66	14.16	53.40	14.19	51.47	14.04	0.001
MBP, (mmHg)	101.89	11.21	103.23	11.17	99.74	10.95	<0.001
Hypertension, n (%)	1795	(72.5)	1172	(76.9)	623	(65.5)	<0.001
Antihypertensive drugs, n (%)	1272	(51.4)	786	(50.4)	504	(53.0)	0.215
Total cholesterol, (mg/dL)	225.53	41.07	220.39	38.92	233.77	43.05	<0.001
LDL cholesterol, (mg/dL)	139.88	34.97	138.34	34.26	142.32	35.96	0.006
HDL cholesterol, (mg/dL)	49.81	13.03	47.89	11.96	52.90	14.04	<0.001
Triglycerides, (mg/dl)	146.21	96.48	150.99	105.90	138.59	78.63	0.001
No-HDL cholesterol, (mg/dL)	175.74	40.74	172.54	38.34	180.86	43.84	<0.001
Atherogenic index, (mg/dL)	4.76	1.31	4.82	1.30	4.67	1.34	0.004
Dyslipidemia, n (%)	1664	(67.2)	969	(63.6)	695	(73.2)	<0.001
Lipid–lowering drugs, n (%)	717	(29.0)	419	(27.5)	298	(31.3)	0.045
FPG, (mg/dL)	107.98	34.83	107.76	33.90	108.35	36.29	0.683
HbA1c, (%)	6.12	1.18	6.06	1.12	6.21	1.26	0.002
Diabetes mellitus, n (%)	842	(34.0)	493	(32.3)	349	(36.7)	0.029
Hypoglycaemic drugs, n (%)	511	(20.6)	289	(19.0)	222	(23.3)	0.009
Height, (cm)	164.56	9.27	169.69	6.78	156.34	6.36	<0.001
Weight, (kg)	79.41	14.67	83.81	13.52	72.36	13.67	<0.001
WC, (cm)	100.95	11.68	102.94	10.52	97.76	12.69	<0.001
BMI, (kg/m^2^)	29.26	4.52	29.06	3.95	29.59	5.28	0.004
Obesity, n (%)	897	(26.2)	510	(33.5)	387	(40.7)	<0.001
Abdominal obesity, n (%)	1546	(62.8)	797	(52.6)	749	(79.3)	<0.001
Plasma creatine, (mg/dL)	0.85	0.23	0.94	0.24	0.71	0.13	<0.001
GFR, (mL/min/1,73 m^2^)	87.47	14.21	86.79	14.87	88.56	13.02	0.003
CVR SCORE scale, (%)	3.55	2.73	4.48	2.93	2.05	1.40	<0.001
**Cardiovascular diseases**							
Renal disease, n (%)	2	(0.1)	2	(0.1)	0	0.00	0.526
Peripheral arteriopathy, n (%)	13	(0.5)	11	(0.7)	2	(0.2)	0.150
Heart failure, n (%)	12	(0.5)	11	(0.7)	1	(0.1)	0.036
**Vascular function**							
baPWV, (m/s)	14.87	2.63	14.82	2.65	14.93	2.60	0.313
baPWV yes Adherence MD, (m/s)	14.80	2.49	14.70	2.48	14.95	2.52	0.068
baPWV non-Adherence MD, (m/s)	14.93	2.78	14.96	2.83	14.90	2.69	0.760
baPWV yes Adherence DQI (m/s)	14.77	2.71	14.74	2.70	14.85	2.72	0.752
baPWV non-Adherence DQI, (m/s)	14.94	2.57	15.08	2.52	14.78	2.71	0.114

Continuous variables are shown as mean ± standard deviation. The categorical variables are shown as a number and percentage. Risk alcohol consumption in women and men was ≥140 and ≥210 g/week, respectively. Individuals were considered sedentary if the moderate physical activity performed was <675 MET-minutes/week or the intense physical activity was <420 MET-minutes/week. Definitions are from the American Heart Association, 2007. Adherence MD ≥ 5. Adherence DQI ≥ 39. N: number; SD: standard deviation; gr/W: grams/week; FA: physical activity; METs/m/W: basal metabolic rate/minute/week; MD: Mediterranean diet; DQI: diet quality index; SBP: systolic blood pressure; DBP: diastolic blood pressure; BP: pulse pressure; MBP: mean blood pressure; LDL: low–density lipoprotein; HDL: high–density lipoprotein; FPG: fasting plasma glucosa; HbA1c: glycosylated hemoglobin; WC: waist circumference; BMI: body mass index; CVR: cardiovascular risk; GFR: glomerular filtration; baPWV: Brachial-Ankle pulse wave velocity. *p* value: differences between men and women.

**Table 2 nutrients-14-00244-t002:** General characteristics of the subjects included by vascular ageing.

	Without EVA	(2210)	With EVA	(265)	*p* Value
Lifestyles	Mean or N	SD or (%)	Mean or N	SD or (%)	
Years of smoker, (years)	31.60	12.70	32.27	14.02	0.555
Smoker, n (%)	652	(29.5)	58	(21.9)	0.010
Alcohol, (gr/W)	70.75	111.72	81.28	152.23	0.276
Risk consumption, n (%)	296	(13.4)	38	(14.3)	0.636
Total FA, (METs/m/W)	2461	2449	2464	2861	0.986
Sedentary, n (%)	1893	(85.7)	234	(88.9)	0.263
MD, (total score)	5.20	1.721	4.98	1.771	0.056
Adherence MD, n (%)	1172	(53.0)	123	(46.4)	0.044
DQI, (total score)	38.78	3.04	38.11	3.21	0.001
Adherence DQI, n (%)	1233	(55.8)	125	(47.2)	0.009
**Conventional Risk Factors**					
Age, (years)	61.24	7.737	62.13	7.397	0.076
SBP, (mmHg)	135.57	16.15	151.27	20.57	<0.001
DBP, (mmHg)	84.00	9.92	89.40	11.42	<0.001
BP, (mmHg)	51.56	13.38	61.88	16.86	<0.001
MBP, (mmHg)	100.94	10.62	109.77	12.83	<0.001
Hypertension, n (%)	1554	(70.3)	241	(90.9)	<0.001
Antihypertensive drugs, n (%)	1113	(50.4)	159	(60.0)	0.003
Total cholesterol, (mg/dL)	226.20	41.03	219.96	40.99	0.019
LDL cholesterol, (mg/dL)	140.78	34.92	132.35	34.61	<0.001
HDL cholesterol, (mg/dL)	49.80	12.93	49.88	13.80	0.928
Triglycerides, (mg/dL)	144.43	91.44	161.23	130.80	0.008
No-HDL cholesterol, (mg/dL)	176.42	40.68	170.08	40.84	0.017
Atherogenic index, (mg/dL)	4.78	1.32	4.63	1.27	0.090
Dyslipidemia, n (%)	1496	(67.7)	168	(63.4)	0.166
Lipid–lowering drugs, n (%)	635	(28.7)	82	(30.9)	0.474
FPG, (mg/dL)	121.04	42.47	106.42	33.474	<0.001
HbA1c, (%)	6.06	1.14	6.59	1.36961	<0.001
Diabetes mellitus, n (%)	710	(32.1)	132	(49.8)	<0.001
Hypoglycaemic drugs, n (%)	413	(18.7)	98	(37.0)	<0.001
Height, (cm)	164.64	9.240	163.86	9.545	0.0198
Weight, (kg)	79.50	14.72	78.61	14.29	0.349
WC, (cm)	100.87	11.78	101.60	10.80	0.340
BMI, (kg/m^2^)	29.27	4.53	29.21	4.37	0.848
Obesity, n (%)	800	(36.2)	97	(36.6)	0.893
Abdominal obesity, n (%)	1371	(62.5)	175	(66.0)	0.256
Plasma creatine, (mg/dL)	0.85	0.19	0.89	0.4168872	0.004
GFR, (mL/min/1,73 m^2^)	87.63	13.85	86.13	16.86132	0.104
CVR SCORE scale, (%)	3.41	2.59	4.66	3.48391	<0.001
**Cardiovascular diseases**					
Renal disease, n (%)	0	(0.0)	2	(0.8)	<0.001
Peripheral arteriopathy	0	(0.0)	13	(4.9)	<0.001
Heart failure	0	(0.0)	12	(4.5)	<0.001
**Vascular function**					
baPWV, (m/s)	14.32	1.93	19.46	3.21	<0.001

Continuous variables are shown as mean ± standard deviation. The categorical variables are shown as number and percentage. Risk alcohol consumption in women and men was ≥140 and ≥210 g/week, respectively. Individuals were considered sedentary if the moderate physical activity performed was <675 MET-minutes/week or the intense physical activity was <420 MET-minutes/week. Definitions are from the American Heart Association, 2007. Adherence MD ≥ 5. Adherence DQI ≥ 39. N: number; SD: standard deviation; gr/W: grams/week; FA: physical activity; METs/m/W: basal metabolic rate/minute/week; MD: Mediterranean diet; DQI: diet quality index; SBP: systolic blood pressure; DBP: diastolic blood pressure; BP: pulse pressure; MBP: mean blood pressure; LDL: low–density lipoprotein; HDL: high–density lipoprotein; FPG: fasting plasma glucosa; HbA1c: glycosylated hemoglobin; WC: waist circumference; BMI: body mass index; CVR: cardiovascular risk; GFR: glomerular filtration; baPWV: Brachial-Ankle pulse wave velocity. *p* value: differences between men and women.

**Table 3 nutrients-14-00244-t003:** Multiple regression analysis of arterial stiffness with diet quality index and Mediterranean diet.

	β	95% CI	*p* Value
DQI			
Total	−0.081	(−0.105 to −0.028)	0.001
Men	−0.118	(−0.145 to −0.054)	<0.001
Women	0.067	(−0.020 to 0.118)	0.162
MD			
Total	−0.052	(−0.141 to −0.008)	0.027
Men	−0.081	(−0.198 to −0.040)	0.003
Women	0.066	(−0.036 to 0.204)	0.168

Multiple regression analysis using baPWV m/s as a dependent variable, DQI and MD scores as independent variables, and age, sex, years as a smoker, alcohol consumption in grams/week, physical activity in METs/m/W, and hypotensive, hypoglycemic, or lipid-lowering drugs as adjustment variables. DQI: diet quality index; MD: Mediterranean diet; baPWV: Brachial-Ankle pulse wave velocity.

## Data Availability

The variables that we have used in the analyses carried out to obtain the results of this work are available upon reasoned request to the corresponding author.

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
