# Peer review of "The Association of Dietary Intake with Arterial Stiffness and Vascular Ageing in a Population with Intermediate Cardiovascular Risk—A MARK Study"

_nutrients, 2022, doi:10.3390/nu14020244_

Round 1

Reviewer 1 Report

The authors describe their work on the association of diet with arterial stiffness and vascular ageing in a Caucasian population with intermediate cardiovascular risk. The results of this study suggest that a greater score in the DQI and MD adherence questionnaires is associated with lower arterial stiffness and lower probability of presenting EVA. In the analysis by sex, this association is only observed in men. This is an interesting study. Appropriate methodology has been employed and the conclusions appear to be justified based on the data at hand. The manuscript is written very well. I have only minor recommendations for consideration.

  1. Introduction. Can the authors provide a clear hypothesis to be tested in the study.
  2. Discussion. While the authors have adequately described the limitations of their study, I would still like to see some discussion on possible mechanisms that account for the sex differences in the observations.
  3. Discussion. Can the authors elaborate and emphasize the clinical applicability of the findings.

Author Response

Comments and Suggestions for Authors

The authors describe their work on the association of diet with arterial stiffness and vascular ageing in a Caucasian population with intermediate cardiovascular risk. The results of this study suggest that a greater score in the DQI and MD adherence questionnaires is associated with lower arterial stiffness and lower probability of presenting EVA. In the analysis by sex, this association is only observed in men. This is an interesting study. Appropriate methodology has been employed and the conclusions appear to be justified based on the data at hand. The manuscript is written very well. I have only minor recommendations for consideration.

  1. Introduction. Can the authors provide a clear hypothesis to be tested in the study.

Authors' Answer

We have added the following sentence in the introduction:

The hypothesis of the study is that subjects with lower adherence to the DQI and MD questionnaires will have higher arterial stiffness, and a higher percentage of them will be classified as EVA.

  1. Discussion. While the authors have adequately described the limitations of their study, I would still like to see some discussion on possible mechanisms that account for the sex differences in the observations.

Authors' Answer

We have added the following paragraph to the discussion, in the section analysing sex differences.

On the other hand, vascular ageing is a gradual process that involves biochemical, enzymatic, and cellular phenomena in the vascular wall. All this, combined with epigenetic and molecular alterations, causes an increase in arterial stiffness and a reduction in compliance that differs according to sex [1,2]. Therefore, sex differences are influenced by hormonal and non-hormonal factors [3,4]. Thus, the protection of endogenous estrogen until menopause in women is well known. In addition, in males, arterial stiffness increases linearly from puberty, indicating that females inherently have stiffer main arteries than males; these effects are mitigated by sex steroids during reproductive life. Height, body fat distribution, and inflammatory factors may also play a role [3,4].

  1. Discussion. Can the authors elaborate and emphasize the clinical applicability of the findings.

Authors' Answer

We have added the following sentence in the discussion at the end of the conclusion:

These results may be useful in clinical practice, helping health professionals to provide healthy dietary recommendations, with the aim of improving vascular ageing, especially in men.

References

  1. Laurent, S.; Boutouyrie, P.; Cunha, P.G.; Lacolley, P.; Nilsson, P.M. Concept of Extremes in Vascular Aging. Hypertension 2019, 74, 218-228, doi:10.1161/hypertensionaha.119.12655.
  2. Cunha, P.G.; Boutouyrie, P.; Nilsson, P.M.; Laurent, S. Early Vascular Ageing (EVA): Definitions and Clinical Applicability. Curr Hypertens Rev 2017, 13, 8-15, doi:10.2174/1573402113666170413094319.
  3. Kane, A.E.; Howlett, S.E. Differences in Cardiovascular Aging in Men and Women. Adv Exp Med Biol 2018, 1065, 389-411, doi:10.1007/978-3-319-77932-4_25.
  4. Rossi, P.; Francès, Y.; Kingwell, B.A.; Ahimastos, A.A. Gender differences in artery wall biomechanical properties throughout life. J Hypertens 2011, 29, 1023-1033, doi:10.1097/HJH.0b013e328344da5e.

Reviewer 2 Report

In the present paper, Leticia Gómez-Sánchez and colleagues analysed the association of diet with arterial stiffness and vascular ageing in a Caucasian population with intermediate cardiovascular risk. The authors concluded that the results of this study suggest that a greater score in the Diet Quality Index and Mediterranean Diet adherence questionnaires is associated with lower arterial stiffness and lower probability of presenting Early Vascular Ageing. Moreover, in the analysis by sex, this association is only observed in men.

Overall, I think that the paper is well written, nice, and timely, and it will be of great interest to the readers and researchers, in general. 

I make some suggestions for further improve the quality of the manuscript.

“Oriental diet” (similar to “Mediterranean diet”) is particularly abundant in isoflavones. Recent observational research indicated that higher intake of isoflavones and tofu was associated with a moderately lower risk of developing Coronary Heart Disease. Specifically, one year of treatment with pure genistein ameliorated the cardiac and endothelial function, as well as improved surrogate endpoints associated with risk for diabetes and Cardiovascular Diseases in postmenopausal women with Metabolic Syndrome. Additionally, a recent work by Dinu and colleagues suggested that the “Mediterranean diet had the strongest and most consistent evidence of a beneficial effect on both anthropometric parameters and cardiometabolic risk factors”

Please discuss these intriguing aspects in the revised section of manuscript, considering for your convenience these references:

-Dinu M.; Pagliai G.; Angelino D.; et al. Effects of Popular Diets on Anthropometric and Cardiometabolic Parameters: An Umbrella Review of Meta-Analyses of Randomized Controlled Trials. Adv. Nutr. 2020 Feb 14.

-Ma L.; Liu G.; Ding M.; Zong G.; et al. Isoflavone Intake and the Risk of Coronary Heart Disease in US Men and Women: Results From 3 Prospective Cohort Studies. Circulation. 2020, 141, 1127-1137.

Author Response

Comments and Suggestions for Authors

In the present paper, Leticia Gómez-Sánchez and colleagues analysed the association of diet with arterial stiffness and vascular ageing in a Caucasian population with intermediate cardiovascular risk. The authors concluded that the results of this study suggest that a greater score in the Diet Quality Index and Mediterranean Diet adherence questionnaires is associated with lower arterial stiffness and lower probability of presenting Early Vascular Ageing. Moreover, in the analysis by sex, this association is only observed in men.

Overall, I think that the paper is well written, nice, and timely, and it will be of great interest to the readers and researchers, in general. 

I make some suggestions for further improve the quality of the manuscript.

“Oriental diet” (similar to “Mediterranean diet”) is particularly abundant in isoflavones. Recent observational research indicated that higher intake of isoflavones and tofu was associated with a moderately lower risk of developing Coronary Heart Disease. Specifically, one year of treatment with pure genistein ameliorated the cardiac and endothelial function, as well as improved surrogate endpoints associated with risk for diabetes and Cardiovascular Diseases in postmenopausal women with Metabolic Syndrome. Additionally, a recent work by Dinu and colleagues suggested that the “Mediterranean diet had the strongest and most consistent evidence of a beneficial effect on both anthropometric parameters and cardiometabolic risk factors”

Please discuss these intriguing aspects in the revised section of manuscript, considering for your convenience these references:

-Dinu M.; Pagliai G.; Angelino D.; et al. Effects of Popular Diets on Anthropometric and Cardiometabolic Parameters: An Umbrella Review of Meta-Analyses of Randomized Controlled Trials. Adv. Nutr. 2020 Feb 14.

-Ma L.; Liu G.; Ding M.; Zong G.; et al. Isoflavone Intake and the Risk of Coronary Heart Disease in US Men and Women: Results From 3 Prospective Cohort Studies. Circulation. 2020, 141, 1127-1137.

Authors' Answer

Following the reviewer's indications we have added the following paragraph to the discussion:

The benefits of different heart-healthy dietary patterns were analysed in the Umbrella review of Meta-Analyses of Randomized Controlled Trials carried out by Dinu et al. [1]. In it, they analysed 11 dietary patterns, including mainly subjects who were overweight, obese, or diagnosed with type 2 diabetes mellitus. The strongest and most consistent evidence was found for the Mediterranean dietary pattern. This diet showed a reduction in weight, body mass index, blood pressure, insulin glucose, and HbA1c and an improvement in the lipid profile without evidence of potential adverse effects.

In this line, a study carried out by Ma et al. [2] in three prospective cohorts of US men and women included in the Health Professionals Follow-Up Study and in the NHS (Nurses' Health Study) with more than two decades of follow-up showed that a higher intake of isoflavones was associated with a moderately lower risk of coronary heart disease. Tofu consumption was also significantly and inversely associated with coronary heart disease risk. The favourable association of tofu was more pronounced in young or postmenopausal women, without hormone replacement therapy. These results imply that tofu and other soy products could be incorporated into the diet to facilitate the prevention of ischemic heart disease [2].

References

  1. Dinu, M.; Pagliai, G.; Angelino, D.; Rosi, A.; Dall'Asta, M.; Bresciani, L.; Ferraris, C.; Guglielmetti, M.; Godos, J.; Del Bo, C.; et al. Effects of Popular Diets on Anthropometric and Cardiometabolic Parameters: An Umbrella Review of Meta-Analyses of Randomized Controlled Trials. Adv Nutr 2020, 11, 815-833, doi:10.1093/advances/nmaa006.
  2. Ma, L.; Liu, G.; Ding, M.; Zong, G.; Hu, F.B.; Willett, W.C.; Rimm, E.B.; Manson, J.E.; Sun, Q. Isoflavone Intake and the Risk of Coronary Heart Disease in US Men and Women: Results From 3 Prospective Cohort Studies. Circulation 2020, 141, 1127-1137, doi:10.1161/circulationaha.119.041306.

Reviewer 3 Report

The reviewed manuscript concerns an interesting evaluation of a large population-based group that aims to analyse the association of diet with arterial stiffness and vascular ageing in 2,475 individuals aged 35-75 years with intermediate cardiovascular risk.

Manuscript contains the key elements according to the journal requirements. Methods are comprehensively described. The results are presented clearly in units and including three tables and one figure summarizing the study data.

Please see my comments below as suggestions how the manuscript can be further improved:

- In the section ‘Discussion’, please highlight to a greater extent the following themes:  Implications of this study and Novelty in this study.

- The numbers of references needs to be adjusted in line with the journal’s requirements.

Author Response

Comments and Suggestions for Authors

The reviewed manuscript concerns an interesting evaluation of a large population-based group that aims to analyse the association of diet with arterial stiffness and vascular ageing in 2,475 individuals aged 35-75 years with intermediate cardiovascular risk.

Manuscript contains the key elements according to the journal requirements. Methods are comprehensively described. The results are presented clearly in units and including three tables and one figure summarizing the study data.

Please see my comments below as suggestions how the manuscript can be further improved:

- In the section ‘Discussion’, please highlight to a greater extent the following themes:  Implications of this study and Novelty in this study.

- The numbers of references needs to be adjusted in line with the journal’s requirements.

Authors' Answer

1.-We have added the following sentence in the discussion

The main novelties of this study are as follows: First, it analyses the relationship of the results between two dietary questionnaires with vascular ageing in Caucasian patients with intermediate cardiovascular risk. Second, it shows that subjects with lower adherence to a healthy diet, as assessed by the DQI and MD questionnaires, are more likely to be classified as EVA. Finally, when we analyse the association separately by sex, this is only maintained in men.

2.-We have added the following sentence in the discussion at the end of the conclusion:

These results may be useful in clinical practice, helping health professionals to provide healthy dietary recommendations, with the aim of improving vascular ageing, especially in men.

3.-We have reviewed the requirements of the journal Nutrients and have found no limit on the number of references.   If there is a limit, please let us know and we will adapt the number of references.

Round 2

Reviewer 2 Report

Thank you for addressing my comments well. I have no further remarks.